# Governments Should Mandate Tiered Anonymity on Social-Media Platforms to Counter Deepfakes and LLM-Driven Mass Misinformation

## Abstract

This position paper argues that governments should mandate a three-tier anonymity framework on social-media platforms as a reactionary measure prompted by the ease-of-production of deepfakes and large-language-model-driven misinformation. The tiers are determined by a given user's *reach score*: Tier 1 permits full pseudonymity for smaller accounts, preserving everyday privacy; Tier 2 requires private legal-identity linkage for accounts with some influence, reinstating real-world accountability at moderate reach; Tier 3 would require per-post, independent, ML-assisted fact-checking, review for accounts that would traditionally be classed as sources-of-mass-information.

An analysis of Reddit shows volunteer moderators converging on comparable gates – karma thresholds, approval queues, and identity proofs – as audience size increases, demonstrating operational feasibility and social legitimacy. Acknowledging that existing engagement incentives deter voluntary adoption, we outline a regulatory pathway that adapts existing US jurisprudence and recent EU-UK safety statutes to embed reach-proportional identity checks into existing platform tooling, thereby curbing large-scale misinformation while preserving everyday privacy.

## 1 Introduction

**Governments should mandate *tiered anonymity* on social-media platforms to curb the democratic harms of deepfakes and large-language-model–amplified misinformation.** When influence is algorithmically amplified and truth is algorithmically optional, the notion that all online voices should enjoy equal anonymity becomes not a right, but a liability. This position responds to the growing asymmetry between the ease with which synthetic content can shape public discourse and the absence of mechanisms to hold the most influential voices accountable. Generative models now enable anyone to manufacture persuasive audio-visual fabrications at negligible cost, eroding the traditional evidentiary value of sight and sound and fueling the "liar's dividend", the tactic of dismissing inconvenient truths as fakes [1, 2]. Simultaneously, recommender systems amplify attention without regard to veracity, allowing fringe messages to reach millions in minutes.

Online anonymity was originally a shield for ordinary speakers, political dissidents, and vulnerable groups. However, when algorithmic amplification gives a single post the reach of a broadcaster, blanket anonymity becomes a public-safety liability. We therefore argue that *identity obligations should scale with influence*.

Our proposed three-tier model (summarized in Table 1) assigns obligations by *reach* (e.g. a weighted sum of followers, shares, views, etc.). This is further explored in Section 4. Tier 1 preserves full pseudonymity for low-reach accounts; Tier 2 requires a platform-held legal-identity link once

Table 1: Proposed tiered anonymity framework. Tier thresholds are discussed in Section 4, are generally illustrative, and should be calibrated per-platform. A single post that crosses a threshold retroactively elevates the account to the corresponding tier.

| Tier | Typical Accounts | Identity & Friction Obligations |
|---|---|---|
| 1 | Personal diaries, hobby groups | Full pseudonymity; no legal-identity linkage. Content governed only by ordinary community rules. |
| 2 | Niche influencers, local news pages | Platform-held verification of a government identity. Cooling-off window for posts; tamper-proof audit log retained. No public disclosure of real-world identity. |
| 3 | National media brands, celebrities | Independent, ML-assisted fact-checking and provenance watermarking *before* algorithmic amplification; searchable public archive of corrections. Non-compliance triggers down-ranking or removal. |

a predefined reach threshold is crossed; Tier 3 adds independent, ML-assisted fact-checking for mass-reach content.

We employ *friction* – any deliberate cost or delay imposed on posting or sharing – as a design principle. This has been shown to reduce misinformation and abusive speech by prompting deliberation. Empirical studies and industry roll-outs of "read-before-retweet" or "reconsider reply" prompts cut harmful interactions and friction, in a social media context, significantly improves the average quality of posts [3–5]. A tiered anonymity regime institutionalizes friction proportionally: identity verification and fact-checking occur only when content exceeds influence thresholds, preserving low-stakes spontaneity while dampening high-stakes manipulation.

We ground our proposal in using empirical evidence from Reddit's community moderation approach. Volunteer moderators already converge on proportional governance: as subreddit traffic grows, moderators introduce karma[1] minimums, manual reviews, or identity checks before posts appear [6–8]. These organic practices demonstrate operational feasibility and indicate latent demand for tiered accountability that transcends any single platform architecture.

We further show that there is a viable regulatory pathway to achieving our proposal. The European Union's Digital Services Act already requires marketplaces to verify business users and offers a blueprint for identity-linked content duties [9]. The UK Online Safety Act obliges "Category 1"[2] services to give adults tools for filtering anonymous accounts and mandates that platforms offer identity verification [10]. By drawing on these precedents, legislators can embed tiered anonymity into safety models and ranking systems without prohibiting pseudonymity outright.

**Contributions**   This paper makes two main contributions:

1. We introduce a formal model that maps user reach to escalating identity and verification duties, capturing both follower-heavy and suddenly viral accounts. This is supplemented by an empirical evidence from a longitudinal Reddit case study that proportional identity governance emerges endogenously in large online communities.

2. We chart a concrete, jurisdiction-spanning regulatory pathway that leverages existing DSA and Online Safety Act provisions to operationalize the model.

By calibrating identity obligations to influence, tiered anonymity restores proportionate friction to digital speech, aligns platform incentives with democratic values, and closes the accountability gap that AI-augmented misinformation eagerly exploits.

---

[1]In Reddit's context, this denotes an aggregate reputation metric equal to the net difference between positive and negative votes that a user's submissions and comments receive. It thus functions as a quantifiable proxy for community trust and is frequently employed as an eligibility threshold for posting or moderation privileges.

[2]Typically, major social media platforms

## 2   Friction, Identity, and Accountability

The starting point for any meaningful reform of online anonymity must confront a central tension in liberal democracies: the commitment to free expression versus the need to mitigate its weaponization. Classic accounts of speech rights, from John Stuart Mill to modern First Amendment jurisprudence, treat expression as a public good – presumptively beneficial and self-regulating [11–13]. Yet in algorithmically mediated environments, where virality can be decoupled from both truth and reputation, the foundational assumptions underpinning these traditions begin to unravel.

Friction – in the form of verification, moderation, or traceability – is often framed as a threat to openness [5]. That said, friction is a *democratic design feature* [14]. In physical communities, social friction arises from reputational consequences, geographic co-presence, and mutual visibility. One is less likely to spread inflammatory falsehoods in a town hall than online, not because one is more moral, but because the social costs are real and immediate. Digital platforms, in contrast, systematically dissolve these frictions. Recommender systems prioritize engagement, not deliberation; speed trumps reflection; and pseudonymity attenuates accountability [15].

This breakdown of reputational checks facilitates what some scholars call "context collapse" – the dislocation of speech from relational context [16, 17]. A user with ten followers may be algorithmically amplified to ten million others without any change in content quality, intent, or reliability. However, the legal system continues to treat both speakers as functionally identical. This is the core problem: the law protects anonymity *symmetrically*, while platforms distribute influence *asymmetrically* [18, 19].

We argue that identity obligations must scale with content reach. This is not a blanket call for real-name policies, which have been rightly criticized for silencing vulnerable speakers [20, 21]. Instead, it is a call for *proportional identity calibration* [22], wherein pseudonymity is preserved for low-reach users, while higher-tier actors must submit to private identity verification and, ultimately, to structured content review [23]. This approach mirrors how democratic institutions already manage power: with increasing transparency and accountability as influence grows [24].

Our Reddit case study, described in Section 3, illustrates this principle in practice. As subreddits expand in size and influence, moderation architectures evolve from permissive to hierarchical: identity checks, posting restrictions, and content approvals become the norm. These organically emergent structures reflect a collective intuition: that scale demands scrutiny, and visibility must be earned.

## 3   Reddit Case Study in Community Moderation at Scale

Reddit offers a 20-year natural experiment in large-scale, bottom-up governance. More than $100\,000$ active communities (subreddits) are overseen by roughly $60\,000$ volunteer moderators who outnumber the platform's $\sim$400 paid administrators by two orders of magnitude [25, 26]. In the first half of 2024 alone users generated 5.33 billion pieces of content; moderators and admins removed 3.1% of it – half by volunteers, 71% of whose actions were automated by tools such as `AutoModerator` [27]. Unpaid labor on this scale has been valued at $3.4 million per year [28].

**Multi-layer Moderation**   Governance operates on three nested layers: (i) site-wide rules enforced by a small admin team, (ii) subreddit-specific rules defined and enforced by volunteer moderators, and (iii) crowd signals (voting, reporting) supplied by ordinary users. Empirical analyses show that popular subreddits add *more* and *stricter* rules as audience size grows, often introducing karma thresholds, URL whitelists, or manual approval queues [6, 8]. High-visibility communities even demand identity proofs: r/BlackPeopleTwitter, for example, required photographic skin-tone verification to curb impersonation [7]. These organically emerging "tiered" signals parallel our proposed reach-based anonymity model.

**Adaptive Structure**   Reddit's structure evolves with scale and external pressure. In 2015, subreddits controlling much of Reddit's front page shut down ("AMAgeddon") to protest inadequate mod tooling, prompting the company to invest in logs, modmail, and automated filters [29]. In 2023 more than $7\,000$ subreddits went private to oppose new API fees, again demonstrating the collective leverage of volunteer governance [30]. Despite these confrontations, the core design – local autonomy constrained by platform-level minima – has remained intact and resilient.

Table 2: Identity and moderation norms on major social-media platforms as of May 2025. "Tiered" denotes any mechanism in which obligations or scrutiny escalate with audience size or monetization status.

| Platform | Community Moderation | Tier-like content checks |
|---|---|---|
| Reddit | Volunteer moderators | Karma / account-age gates `AutoModerator` keyword filters Stricter rules as subreddit size grows |
| Facebook | No | Centralised review by staff and contractors No escalation tied to reach |
| Instagram | No | Feature gates at $\sim$10k followers (links, product tags) Content demotion or removal on policy breach |
| X (Twitter) | Community Notes | No systematic reach-based review Enforcement tied to policy breaches |
| TikTok | No | Increased human review for high-follower creators Scaled ML enforcement for long-tail users |
| YouTube | No | Automated checks for new channels Manual review for Partner-Program content ($\geq$ 1k subs) Additional scrutiny for 100k+ channels |

**Scale causes Friction**    Quantitative work finds a positive correlation between subreddit size and the likelihood of (i) entry gates (minimum account age/karma) [8], (ii) pre-publication queues [8, 31], and (iii) ex post identity checks [7]. In other words, moderators intuitively impose *proportional friction*: low-reach users post freely; higher-reach content encounters verification or review. Reddit thus supplies real-world evidence that tiered anonymity is operationally feasible and socially accepted when the costs of influence are borne chiefly by those who wield it.

**Contrast with Centralized Platforms**    Competing platforms provide no comparable venue for community-level rule-making. Facebook real-name enforcement, X's paid "blue check", and YouTube's purely algorithmic filters all exemplify top-down moderation with minimal local discretion or tiering. Comparative studies confirm that Reddit alone relies "more or less on self-moderation by volunteers", producing a distinctive, multi-layer oversight regime [32]. We summarize our findings regarding identity and moderation norms in all current major social media platforms in Table 2. YouTube provides the closest analogue to our tiered system: "new" channels face automated checks, Partner-Program creators add identity and monetization audits, and six-figure-subscriber channels receive further manual review and provenance badges.

**Take-Away**    Reddit's layered system demonstrates that *identity obligations can scale with reach* without eroding baseline pseudonymity. The empirical pattern – stricter gates as audiences expand – mirrors the normative logic of our three-tier framework and supplies a practicable blueprint for regulatory codification on platforms that lack subreddit-style boundaries.

## 4 Proposed High-level Technical Implementation

Most large platforms already store granular engagement telemetry (followers, impressions, reshares, watch-time). A platform-side service can aggregate these signals into a rolling *reach score* and map it to the tier thresholds proposed in Table 1. To avoid the effect of one-off viral spikes, thresholds should be evaluated over some time period, say a three-month moving window, and updated nightly. When a score first crosses a threshold, a workflow flags the account for *tier elevation* and temporarily rate-limits outbound posts until the verification step – ID upload for Tier 2; fact-checking for Tier 3 – is completed.

Platforms should complement hard metrics with contextual triggers such as monetization enrollment or activation of business tools. Precedent exists: Instagram withholds external–link "Swipe-Up" stories until an account reaches $\sim 10\,000$ followers or holds a business profile, effectively coupling functionality to influence [33]. A similar gating mechanism can enforce tier promotion automatically while minimizing false positives.

**User-facing Controls**   The UK Online Safety Act 2023 obliges Category 1 services to provide adults with filters that exclude non-verified users [10]. A tiered system can generalize this idea: clients expose a preference pane that lets users down-rank or hide Tier 1 content, surface fact-check banners for Tier 3 posts, or receive warnings when resharing material from unverified sources. Such controls translate legal duties into actionable UX.

**Tier Details**   We provide some illustrative guidelines for our proposed tiers:

*Tier 1* No additional obligations: posts remain subject only to baseline community rules.

*Tier 2* Accounts must complete *private* identity verification and comply with advertising-law disclosure. The US FTC's *Endorsement Guides* require influencers to reveal any "material connection" with brands in a manner that is "clear and conspicuous" [34]. Automated classifiers can flag suspected undisclosed ads for moderator review.

*Tier 3* High-reach accounts are treated as de-facto publishers. Posts containing political, health, or financial claims are routed – before wide distribution – to an external fact-checking queue. Empirical surveys by UNESCO show that 62 % of digital creators do not verify information before sharing, underscoring the need for mandated review [35]. Provenance watermarks and a public correction log close the feedback loop; serious or repeated violations trigger algorithmic down-ranking or suspension.

**Progressive Friction**   Existing platform tooling provides technical backing to ensure the necessary friction is applied:

- *Rate-limited publishing queues* that lengthen with tier: seconds for Tier 1, minutes for Tier 2 (cool-off), hours for Tier 3 pending fact-check.
- *Priority triage* of user reports: complaints about Tier 3 content land at the top of moderator dashboards.
- *Automated provenance signals* (e.g. C2PA hashes [36]) injected at upload time for Tier 3 media, enabling rapid debunking should manipulations surface.

These mechanisms impose costs proportionate to communicative power while leaving ordinary pseudonymous speech largely untouched, thereby operationalizing the normative principle that *influence entails accountability*.

## 5   Current Legal Precedents and Regulatory Infrastructure

### 5.1   European Union: From KYBC to Reach-Based Identity Accountability

The European Union provides the strongest foundation for codifying tiered identity obligations. The Digital Services Act (DSA) already introduces structural mechanisms that can be repurposed to support a reach-based verification regime. Article 30's *Know Your Business Customer* (KYBC) requirement, which mandates identity verification for commercial users, represents a conceptual shift: platform functionality is increasingly conditioned on user transparency [37].

More significantly, Articles 34 and 35 impose systemic risk obligations on Very Large Online Platforms (VLOPs) – defined by a monthly audience threshold – explicitly linking reach to responsibility [38]. This sets a critical precedent: the broader a user or platform's influence, the greater the required diligence. Article 9 further enables identity disclosure in response to illegality, reinforcing a principle of proportionality that mirrors the core logic of tiered anonymity [39].

Moreover, complementary frameworks like the AI Act and proposed AI Liability Directive further strengthen this trajectory. By requiring labeling of synthetic media and audit trails for AI systems, the

EU is already enforcing traceability in high-risk communicative environments [40, 41]. A Tier 3 user framework – where mass-reach actors are required to verify identity, disclose sponsored content, and submit to fact-checking – fits squarely within this expanding digital *acquis*. These instruments, taken together, suggest that scalable identity obligations based on content reach are not only compatible with EU law – they are its logical extension.

## 5.2 United Kingdom: The Online Safety Act and Voluntary Verification

By contrast, the UK's Online Safety Act 2023 [10] establishes a statutory duty of care on digital platforms, particularly those classified as Category 1 services – platforms with significant reach and functionality. Under the accompanying Categorization of Regulated Services Threshold Conditions Regulations 2024 [42], these platforms are required to offer adult users the option to verify their identity and to provide tools enabling content filtering based on verification status. This framework introduces a layered reputational infrastructure while preserving the right to anonymity, laying the conceptual groundwork for a tiered identity regime.

However, this identity framework remains voluntary and reputational rather than mandatory and enforceable. Users may choose to verify themselves, and others may opt to filter content accordingly – but no binding obligations are imposed on high-reach users who remain anonymous. Legal mechanisms such as the Norwich Pharmacal orders and the UK–US CLOUD Act [43, 44] already allow for identity disclosure under judicial or governmental request, affirming that anonymity online is not absolute but subject to contextual limits.

Nevertheless, the current UK regulatory landscape lacks a proactive mechanism linking user influence – measured by visibility, engagement, or monetization – to identity obligations. We argue that this omission is increasingly untenable in an era of algorithmic virality, where individuals can rapidly attain significant reach with little to no accountability.

A logical evolution of the Online Safety Act would be to mandate identity verification for users who exceed a defined influence threshold. This threshold could be determined through transparent metrics such as sustained follower counts, average post reach, or eligibility for monetization tools. Such a reform would convert identity verification from a reputational indicator into a mechanism of enforceable accountability.

By embedding this obligation within the existing statutory framework, the UK could pioneer a *rights-preserving* yet *responsibility-tiered model* of online governance – one that maintains anonymity for everyday users while ensuring that high-reach actors meet proportionate standards of transparency and legal traceability.

## 5.3 United States: First-Amendment Boundaries and Conditional Immunity

The United States presents the most challenging jurisdiction for any form of compelled identity regulation due to robust First Amendment protections and the shield of Section 230 of the Communications Decency Act [45]. American courts have repeatedly upheld the right to anonymous speech, particularly in digital spaces. Landmark cases such as *Doe v. Cahill* and *Dendrite Int'l, Inc. v. Doe No. 3* [46] require plaintiffs seeking to unmask anonymous users to meet stringent standards, such as presenting a *prima facie* case of harm and passing a balancing test that weighs the speaker's right to anonymity.

Despite this, momentum is growing at the federal level toward rethinking the blanket nature of Section 230 immunity. Legislative proposals – including bipartisan efforts – have increasingly considered conditioning immunity on a platform's compliance with transparency and good-faith content moderation practices [47, 48]. Rather than mandating identity disclosure, these proposals suggest a path for indirect, incentive-based regulation that respects constitutional limits while introducing mechanisms of accountability.

In this context, the tiered identity framework proposed in Section 3 offers a legally viable and technically feasible approach. Platforms could retain full Section 230 protections only if they adopt a structured system of user obligations based on influence. Such a framework would allow users to remain anonymous at lower tiers but require incremental disclosures or review processes as their reach – and thus potential for public impact – increases. For example, Tier 2 accounts would undergo

private identity verification, while Tier 3 accounts would trigger pre-distribution fact-checking for sensitive content and incorporate provenance watermarks such as C2PA hashes [36].

This model introduces calibrated friction aligned with communicative power. Publishing latency, complaint prioritization, and enhanced moderation protocols ensure that higher influence comes with proportionate responsibility. Importantly, these obligations are not imposed by fiat, but rather tied to platform-side metrics such as engagement telemetry and monetization enrollment. This allows the system to remain content-neutral and voluntary, which is crucial for surviving constitutional scrutiny [49, 50].

Furthermore, this model dovetails with user-choice provisions already emerging in US and UK law. For instance, adults using major platforms under the UK's Online Safety Act 2023 can opt to filter out unverified users [10]. US platforms could offer analogous controls – such as the ability to down-rank Tier 1 content or flag Tier 3 posts with fact-check banners – thus translating normative goals into tangible UX affordances.

In sum, a tiered framework based on influence rather than identity per se provides a constitutionally sound middle ground. It operationalizes the principle that "influence entails accountability", not by restricting speech, but by assigning procedural obligations where amplification is algorithmically enabled [51].

# 6 Piercing Anonymity and Legal Thresholds

While previous sections have outlined the legal mechanisms available to unmask anonymous actors, this section turns from retrospective tools to the conceptual and operational implications of prospective identity collection – that is, requiring platforms to obtain verifiable identity data from users before harms occur, based on the scale of their content reach.

Legal regimes in the EU, UK, and US all permit ex post identity disclosure in narrowly defined circumstances. Yet these mechanisms often prove too slow or reactive for mitigating fast-moving misinformation. Courts and regulators typically intervene only after content has already spread and caused damage – by which point the harm is often irreversible [52, 53]. Moreover, these frameworks do not scale well in a high-volume, high-speed platform environment.

The tiered anonymity model proposed here shifts this paradigm. For Tier 2 and Tier 3 users – those with moderate to large followings – platforms would be required to collect and securely store legal identity information *in advance*, subject to minimal access protocols and stringent privacy protections. This would allow for swift disclosure upon valid legal request while protecting pseudonymity in everyday use. The goal is not to reduce anonymity universally, but to *contextualize it based on communicative power* [54, 55].

Crucially, this shift does not necessitate the rewriting of existing legal thresholds for unmasking identities. Rather, it enhances procedural efficiency and evidentiary readiness when those thresholds are met. For example, a court order that might normally take weeks to execute due to jurisdictional barriers and technical resistance could be processed swiftly if the platform has already verified identity and established a lawful disclosure protocol [20].

To preserve civil liberties, identity databases must be governed by robust safeguards. These include:

- **End-to-end encryption** for stored identity data
- **Access logging** to track who requests and receives information
- **Data minimization** (collecting only what is necessary)
- **Retention limits** with periodic review and deletion
- **Cross-border legal harmonization**, particularly through MLATs and agreements like the CLOUD Act

This approach reframes identity not as a binary attribute, but as a *regulated credential* – conditionally disclosed, proportionately applied, and safeguarded by due process. As such, it avoids the pitfalls of South Korea's real name policy while addressing the increasing costs of untraceable amplification [56]. Ultimately, prospective identity collection enables responsiveness without repression – a legal architecture suited for the velocity and asymmetry of the contemporary information ecosystem.

# 7 The Global Momentum for Conditional Pseudonymity

The international policy environment is increasingly converging around the idea that identity obligations should scale with user influence. Early efforts to regulate anonymity, such as South Korea's real-name verification law (2009–2012), sought to impose identity disclosure universally. That approach proved both legally unsustainable and practically ineffective. The Korean Constitutional Court invalidated the policy for violating freedom of expression, and subsequent research showed it failed to reduce online harms in any measurable way [56, 57]. The lesson was clear: blanket identity mandates are blunt instruments that overreach without precision.

Since then, regulatory energy has shifted toward more granular, influence-sensitive models. In India, the 2023 Draft Digital India Bill introduces a risk-based classification framework for digital intermediaries, suggesting a shift toward more nuanced regulatory obligations based on the type and scale of service – but without explicitly extending these obligations to individual users or calibrating them to user influence [58]. Australia's eSafety Commissioner has advanced similar proposals, calling for the traceability of high impact accounts, particularly those linked to harmful or AI-generated content [59]. Meanwhile, the European Commission has initiated consultations on "influence transparency", exploring how verification requirements might apply to accounts disseminating politically sensitive or synthetic media [9, 60]

Platform ecosystems increasingly reflect this logic, though in a fragmented manner. Meta's Verified program, X's (formerly Twitter) "blue check" system, and YouTube's monetization criteria all condition algorithmic reach, visibility, and revenue on voluntary identity disclosure or engagement thresholds [61–63]. These systems reinforce a de facto hierarchy: creators with broader audiences receive preferential treatment – while also facing greater scrutiny – forming an implicit structure of tiered governance. However, these frameworks often lack transparency, consistency, and regulatory oversight [64].

Taken together, these developments suggest the emergence of a normative shift: pseudonymity remains appropriate for ordinary users, but must give way to verification and procedural safeguards – such as identity linkage, content moderation, or algorithmic throttling – once a user's reach crosses a defined threshold. We term this evolving model *conditional pseudonymity*: a regulatory philosophy that preserves privacy for the many while introducing graduated accountability for the influential.

Our proposed three-tier anonymity framework builds on this global momentum. It does not introduce a wholly new system, but rather formalizes a trend already unfolding across jurisdictions and platforms. By codifying conditional pseudonymity, we provide a principled, scalable model rooted in proportionality and procedural fairness. It aligns regulatory tools with the actual distribution of digital power – preserving pseudonymity where appropriate, qualifying it where necessary, and ultimately ensuring that privacy and accountability evolve in tandem in the algorithmic public sphere.

# 8 Cross-Jurisdictional Implementation and Extraterritorial Reach

Implementing a tiered anonymity framework in a globally interconnected internet ecosystem presents significant enforcement challenges. While Reddit shows that moderation hierarchies can emerge organically, its reliance on volunteer governance is difficult to replicate on commercial, transnational platforms like Meta, YouTube, or X. These platforms operate across multiple jurisdictions but often default to the legal norms of their home country – typically the United States – resulting in fragmented regulatory oversight [65].

To scale tiered anonymity, enforcement must be institutional, driven by governments and platforms rather than individual users. Governments in regions such as the EU, UK, and US already exercise regulatory authority over platforms operating within their borders. This authority can be extended extraterritorially, as seen with the EU's General Data Protection Regulation (GDPR), which extends obligations beyond EU borders through data adequacy requirements and reputational enforcement mechanisms [66, 67].

Instead of basing obligations solely on user location, platforms could use geolocation, engagement metrics, or declared jurisdiction to apply higher-tier requirements based on influence. Tier 2 and 3 features – such as monetization or algorithmic amplification – would require identity verification globally. Users unwilling to verify could still post, but without access to amplification tools.

In lower-regulation or infrastructure-poor jurisdictions, implementation could be supported by interoperable digital identity standards that align with GDPR principles of data minimization, purpose limitation, and secure storage. Public-private partnerships or open-source systems, such as the European Digital Identity (EUDI) Wallet or India's Aadhaar infrastructure (with appropriate safeguards), could provide privacy-preserving verification without broad data disclosure [68–70].

However, unilateral regulation risks being seen as digital imperialism, especially in the Global South [71]. To address this legitimacy challenge, multilateral cooperation is essential. Institutions such as the United Nations Internet Governance Forum (IGF), the OECD, and regional organizations like the African Union and ASEAN can serve as venues for aligning policies and establishing shared norms [72, 73]. Soft-law instruments – non-binding principles, technical standards, and voluntary codes of conduct – can serve as transitional tools toward global harmonization [74, 75].

Framing tiered anonymity as a rights-preserving model is essential. It does not eliminate anonymity, but conditions amplification on influence. By applying identity obligations only at high reach levels and protecting vulnerable users – like whistleblowers and journalists – the framework ensures that accountability scales with power, not participation [76].

Still, resistance is inevitable. Platforms may object to the complexity and cost of implementation or worry about user attrition if stricter identity rules push users to fringe platforms. Likewise, some users may attempt to evade tiering by migrating to less-regulated services.

To address this, enforcement must be both staged and strategic. High-leverage jurisdictions like the EU, UK, and US can drive adoption by linking regulatory compliance to market access. App store requirements, advertising standards, and cross-border data flow agreements can reinforce these incentives [74]. Multilateral coordination can ensure that interoperability standards and privacy safeguards are respected, minimizing the risk of regulatory fragmentation [77].

Ultimately, tiered anonymity is not about censoring speech – it is about regulating amplification. By tying verification and procedural obligations to a user's influence rather than their identity alone, this framework safeguards privacy for ordinary users while ensuring that those with outsized reach meet higher standards of accountability [78]. In this way, tiered anonymity supports a more equitable digital ecosystem – balancing privacy, expression, and responsibility in a scalable, democratic way.

# 9 Conclusion

This paper has advanced a single claim: *governments should require social-media platforms to calibrate anonymity to communicative reach*. By analyzing the epistemic harms of deepfakes and LLM-assisted misinformation, we show that the traditional symmetry of online anonymity no longer maps onto the asymmetry of algorithmic amplification. Our three-tier framework operationalizes the principle that *influence entails accountability*: Tier 1 preserves full pseudonymity, Tier 2 introduces private identity linkage, and Tier 3 imposes publisher-level duties of verification and provenance.

The proposal rests on three pillars. First, empirical evidence from Reddit demonstrates that volunteer moderators already impose proportional gates – karma thresholds, approval queues, and identity checks – as audience size grows [8, 6, 7]. Second, we outlined a technically modest implementation that repurposes existing reach telemetry and friction mechanisms such as rate-limited queues and provenance tagging. Third, we traced a viable regulatory pathway: the EU Digital Services Act [9], the UK Online Safety Act [10], and evolving US jurisprudence [79] already link influence to heightened diligence. Tiered anonymity therefore extends, rather than disrupts, the current legal trajectory.

Adopting this model would re-introduce the social friction that recommender systems have eroded, dampening the incentive and impact of large-scale disinformation while sparing ordinary users from onerous disclosure. Future work must refine threshold calibration, explore privacy-preserving credential systems, and evaluate cross-jurisdictional interoperability. Nonetheless, the core insight is robust: *when speech scales to millions, so must responsibility*. Tiered anonymity offers a scalable, rights-respecting mechanism for restoring that balance.

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
