# OpenReview forum: "Governments Should Mandate Tiered Anonymity on Social-Media Platforms to Counter Deepfakes and LLM-Driven Mass Misinformation"
_NeurIPS.cc/2025/Position_Paper_Track — Submitted to NeurIPS 2025 Position Paper Track_

### Official Review · Reviewer_mZTe · 2025-07-31

**Significance:** 3
**Presentation:** 3
**Rating:** 8
**Confidence:** 4

**Summary:**

This position paper argues that governments should mandate a tiered anonymity framework on social media platforms to counter the growing harms of deepfakes and LLM-driven misinformation. The authors propose a three-tier system where identity and verification obligations grow proportional to the user's reach and influence. Here tier 1 preserves pseudonymity for small accounts, tier 2 introduces private identity verification and tier 3 imposes fact-checking before publishing and traceability. The paper uses Reddit moderation practices as empirical evidence, outlines a technical implementation pathway and looks at the regulatory frameworks in the EU and US. The position is that influence should be proportional to accountability and that the proposed tiered anonymity provides a framework to ensure privacy and responsibility in social media discussions.

**Strengths:**

The paper offers a well-written, legally informed proposal for regulating online anonymity in the age of LLMs and deepfakes. It supports the proposed model with empirical case studies e.g. Reddit, feasibility analysis and a detailed regulatory mapping across geographies. The discussion around different regulations is well-structured and shows that the proposal is legally grounded and not hypothetical. The Reddit case study adds credibility how tiered moderation could emerge organically. The paper covers both societal impact and practical feasibility of the proposed framework.

**Weaknesses:**

- The paper mentions "deepfakes" prominently in the title and abstract but appears not to touch upon this specific threat throughout the paper. This undermines the idea of the proposed framework.
- There are multiple nuanced identity concepts used e.g. conditional pseudonymity, proportional friction, identity calibration but lacks a clear taxonomy overview. This may or may not be obvious to the reader. A table contrasting anonymity models and obligations by country or policy context would help improve reader clarity.
- While Reddit provides a useful analogue, the applicability of community-driven governance to global-scale commercial platforms e.g. Meta, X is debatable and may be overstated. In lines 328-330, the claim that such moderation hierarchies are difficult to replicate is asserted without clear justification and source.
- While the legal mapping is thorough, the paper does not directly address the practical and legal challenges of implementing such a framework in the US i.e. on how it would interact with the constitutional protections and potential resistance from commercially motivated platforms, which may pose practical constraints on the adoption.

**Questions:**

- Could the tier system unintentionally deter mid-tier creators from growing their audience due to additional burden or scrutiny?
- The authors assert that Reddit-style governance is difficult to replicate on commercial platforms but do not explain in detail why. What makes Reddit exceptional in this regard? Is it cultural, structural, legal or commercial?
- Have the authors considered how react thresholds may be gamed e.g. using bots or similar?
- Would it be helpful to include a summary table comparing anonymity obligations and legal constraints across different jurisdictions? This may strengthen the position of the paper.

**Alternative Position:**

Yes, and alternative positions are well-considered and addressed by the argument

**Author Identification:**

No.

**Context:**

3

**Discussion:**

4

**Ethics:**

["NO or VERY MINOR ethics concerns only"]

**Position:**

Yes, the paper argues for or against a position related to machine learning.

**Support:**

4

**Thoroughness:**

4

---

### Official Review · Reviewer_jimX · 2025-08-04

**Significance:** 3
**Presentation:** 3
**Rating:** 6
**Confidence:** 4

**Summary:**

This position paper argues governments should mandate a three-tier anonymity framework on social media platforms to counter AI-generated misinformation and deepfakes. The framework scales identity obligations with user reach: Tier 1 preserves full pseudonymity for low-reach accounts, Tier 2 requires private identity verification for moderate-influence users, and Tier 3 mandates independent fact-checking for mass-reach content. The authors use Reddit's community moderation as evidence that tiered governance emerges organically and is operationally feasible. They outline regulatory pathways through existing EU Digital Services Act, UK Online Safety Act, and evolving US jurisprudence to implement reach-proportional identity verification while preserving privacy for ordinary users.

**Strengths:**

The paper tackles a genuinely important problem with a novel, proportionate solution that avoids the extremes of blanket anonymity or universal real-name requirements. The Reddit empirical evidence effectively demonstrates organic emergence of tiered governance. The cross-jurisdictional regulatory analysis is thorough and identifies concrete implementation pathways. The technical framework is detailed yet feasible, building on existing platform capabilities.

**Weaknesses:**

The definition of "reach" and threshold calibration remains underdeveloped - critical implementation details that could determine success or failure. The paper underestimates enforcement challenges, particularly regarding jurisdiction shopping and technical circumvention. The assumption that platforms will comply without significant resistance may be overly optimistic. Limited analysis of how authoritarian regimes might exploit these frameworks for censorship purposes.

**Questions:**

How would you address the fundamental challenge of defining and measuring "reach" consistently across platforms with different engagement models? What specific safeguards would prevent authoritarian governments from exploiting tiered identity requirements for political suppression? How might the framework adapt to emerging platforms or communication technologies that don't fit traditional social media models? What evidence suggests that fact-checking requirements for Tier 3 users would be more effective than current voluntary approaches?

**Alternative Position:**

Yes, and alternative positions are well-considered and addressed by the argument

**Author Identification:**

No.

**Context:**

3

**Discussion:**

3

**Ethics:**

["NO or VERY MINOR ethics concerns only"]

**Position:**

No, the paper argues that a specific technical approach is superior to other approaches.

**Support:**

3

**Thoroughness:**

4

---

### Official Review · Reviewer_7TUL · 2025-08-28

**Significance:** 4
**Presentation:** 4
**Rating:** 6
**Confidence:** 4

**Summary:**

This paper argues for governments to institute a tiered approach to user anonymity for social media platforms that would assign user accounts to one of three tiers, with progressively stricter accountability requirements.  The goal is to reduce the proliferation of deepfakes and misinformation.  The tiers are assigned based on a user's "reach" (sum of followers, shares, views, and other elements of influence).  Observations of Reddit norms are used to support the proposed framework.  The paper also demonstrates how the proposal is consistent with existing legislation in the EU and U.K. and some elements of U.S. legislation.

**Strengths:**

The paper makes a clear argument for the need for tiered anonymization for users of digital platforms.  This is a particularly compelling statement: "Ultimately, tiered anonymity is not about censoring speech – it is about regulating amplification" (line 366). Rather than burying it at the end of section 8, this could be a strong way to lead the reader into the idea (since most first reactions will involve a concern about the suppression of free speech).

The strongest part of the paper is its grounding in legal and regulatory contexts in the EU, U.K., and U.S. (Section 4) and globally (the first part of section 7).

The topic is very relevant to the social media and online platform communities.  It is less clear that it relates strongly to the NeurIPS (machine learning and computational neuroscience) communities.  Still, it is likely to inspire debate and discussion.

Suggestion:
1. It would help to explain to reader what "shield" section 230 provides since not everyone is familiar with it.

**Weaknesses:**

(1) The Reddit evidence is weak. The claim that strict rules naturally occur with increasing subreddit sizes (lines 104-106) should be supported with data relating subreddit size to a measure of "friction". The claim that Reddit’s system scales "without eroding baseline pseudonymity" is unsupported. Lines 107-108: "High-visibility communities even demand identity proofs" which must erode pseudonymity.

(2) The paper does not demonstrate that governments are the right agents to act. Any restriction on speech by the U.S. government invites constitutional challenges, despite the claim that the proposal is "constitutionally sound" (lines 254-255). Platform action or industry standards would avoid that challenge. No such alternatives (or others) are addressed.

Suggestions:

1. Define "pseudonymity".

2. The paper states that it "restores proportionate friction to digital speech" (lines 62-63), but it does not show that there ever was "proportionate friction" in the past, so it cannot be "restored". Online anonymity has been part of the Internet since early on; see e.g. the "on the Internet, nobody knows you're a dog" cartoon from the New Yorker in 1993.

3. The U.S. Take It Down Act (May 2025) relates to Section 5.3 (lines 249-253).

**Questions:**

The scope of this paper seems to be global, encouraging all governments to employ a tiered anonymity system.  Is the intent that all countries would work to identify the same tiered anonymity system (for consistency)?  If not, wouldn't the proposal face the same fragmentation challenges discussed in section 8?  Lines 348-353 seem to dance around this issue by calling for "cooperation" and aiming for "harmonization" but not coming out and stating that the same tiers should be used everywhere.  Therefore the intent is not clear.

**Alternative Position:**

No

**Author Identification:**

No.

**Context:**

4

**Discussion:**

3

**Ethics:**

["NO or VERY MINOR ethics concerns only"]

**Position:**

Yes, the paper argues for or against a position related to machine learning.

**Support:**

3

**Thoroughness:**

4

---

### Note · Authors · 2025-08-30

**1-10 Additional Comments:**

N/A

**1-11 Submit Again:**

Probably yes

**1-1 Submission Process:**

4

**1-2 Next Year:**

In general, it would be nice to see more emphasis on real-world impact and evaluation; this would encourage arguments grounded in practical deployment, both where things have gone right and where unintended consequences emerged. Additionally, there could be a more promotion of societal and ethical policy frameworks; position papers are in a good position to proactively address regulatory proposals, ethical licensing frameworks, etc.

**1-3 Future Development:**

We have a number of ideas we would like to mention. (1) It would be interesting to see if one of the award categories could be a "living position papers", where the authors (or the whole community) could be encouraged to maintain the selected paper as "living documents" with post-conference updates or linked discussions. (2) We would also like to see some stronger incentives for boldness in the discussed positions, this could be via (textual) signalling that controversial, unconventional, or minority perspectives are welcome, even if they go against mainstream trends. (3) Lastly, it might be interesting to have a curated theme(s) each year of what the conference authors consider to be the most pressing issue at the time. This would be alongside open submissions, and the small set of themes would exist to focus discourse and attract concentrated dialogue.

**1-4 Interest:**

["Panel discussions with other position paper authors", "Structured debates on controversial topics"]

**1-4 Other Interest:**

N/A

**1-5 Thoughtful:**

7

**1-6 Supportive:**

8

**1-7 Technical Aspects Versus Position:**

6

**1-8 Gate Keeping:**

8

**1-9 Camera Ready Changes:**

1. Strengthen core position (7TUL): Move "tiered anonymity regulates amplification, not speech" to the intro; add ML touchpoints (recs amplification; ML-assisted provenance/fact-check).
2. Define key terms (7TUL, mZTe, jimX): Add "Definitions": pseudonymity vs anonymity; proportionate friction; formalize reach (rolling, normalized impressions; 3-mo smoothing), thresholds + sensitivity, per-platform mapping
3. Deepfakes (mZTe): New subsection: Tier-3 pre-amp review of synthetic political/health/finance; C2PA at upload; public corrections log; cross-ref AI Act/C2PA.
4. Reddit evidence (7TUL, mZTe): Add a figure linking subreddit size→friction (karma gates, queues); temper claim, note exceptions, narrow to "operational feasibility"; soften "difficult to replicate" and justify.
5. Enforcement & circumvention (jimX, 7TUL, mZTe): "Jurisdiction shopping and evasions": staged roll-out (EU/UK/US), app-store levers, interop; clarify global gating of amp/monetization; safeguards (velocity caps, anomaly detection, reputation decay) + audit/appeal.
6. Safeguards vs authoritarian misuse (jimX): Content-neutral triggers; independent Tier-3 oversight; due-process for identity disclosure; data-min, encryption, retention limits; press/whistleblower carve-outs; stronger privacy.
7. US policy pathway (7TUL, mZTe): Footnote Section 230; tie to conditional immunity; note May 2025 Take It Down Act; expand First-Amendment (content-neutral, voluntary, amplification-focused).
8. Global harmonization (7TUL): Common 3-tier baseline; jurisdiction-specific thresholds/remedies; IGF/OECD soft-law path; avoid "digital imperialism".
9. Tier-3 fact-checking intent (jimX): Target high-risk claims; algorithmic triage; auditability + SLA timeliness; cite evidence; distinguish mandatory pre-distribution gate from current voluntary.
10. Mid-tier creator burden (mZTe): Mitigations: one-time private ID check; grace/appeals; limited cool-offs; low-risk exemptions.

**3-1 Review Response1:**

jimX

**3-2 Reaction To Review1:**

We thank the reviewer for the thoughtful review. We address each point below.
 - Definition/calibration of "reach": We will add a short Definitions section after the Introduction with to address the questions raised. We also clarify "pseudonymity vs. anonymity" and "proportionate friction", and add a small quantitative figure linking subreddit size to friction in Section 3.
 - Enforcement/circumvention: Sections 8 covers cross-jurisdictional implementation and extraterritorial reach. We will add a subsection addressing jurisdiction shopping and evasion, with points such as staged EU/UK/US rollout, app-store levers. We will also discuss velocity caps, anomaly detection for inorganic spikes, reputation decay, and audit/appeal mechanics.
 - Platform resistance: Our legal analysis in Section 5 already relies on conditional pathways (EU DSA duties, UK OSA tools, US conditional-immunity-style incentives). We will soften language about Reddit portability and add citations on replication costs.
 - Safeguards: Section 6 details privacy/due-process measures (encryption, logging, minimization, retention limits, lawful-process disclosure). We will make guardrails explicit with suggestions such as content-neutral triggers, independent oversight for Tier-3 queues, and whistleblower/journalist carve-outs. We'll move "tiered anonymity regulates amplification, not speech" to the start of the Introduction.
 - Tier-3 evidence/intent: Table 1 and Section 4 treat Tier-3 as publisher-level. We will clarify scope (high-risk political/health/finance), triage/auditability, SLA limits for timeliness, and distinguish this mandatory gate from voluntary approaches with added citations.
 - New platforms: Section 4's contextual triggers (e.g., monetization enrollment, business tools) plus telemetry-based reach extend to new modalities; the Definitions section will make per-platform mappings explicit.

**3-3 Review Response2:**

mZTe

**3-4 Reaction To Review2:**

We thank the reviewer for the thoughtful review and the positive notes on our legal grounding, feasibility, and the Reddit case study. We address each point below.
- Scope / deepfakes: We agree the deepfake thread should be more explicit. We'll carry through the thread from Section 1 more consistently. In the camera-ready, we'll add a short subsection tracing how Tier-3 mitigates deepfake harms (triage of political/health/finance claims; provenance at upload; public corrections log) and cross-reference the EU AI Act discussion.
- Terminology / taxonomy: We'll add a brief Definitions section with formal definitions, and a summary table contrasting anonymity/obligation models and legal constraints across jurisdictions.
- Reddit case study / generalizability: We'll soften "difficult to replicate" and narrow the claim to operational feasibility on large communities. We'll add a quantitative figure and brief methods linking community size to friction, and cite structural differences on commercial platforms.
- U.S. practicality: Section 5.3 positions our model as content-neutral and amplification-oriented, with a conditional-immunity pathway. We'll expand with a footnote on the Section 230 "shield", tie to conditional-immunity proposals, emphasize voluntariness and reach-based triggers, and note the May 2025 "Take It Down Act" as related work on traceability/identity.

> Could Tier-2 deter mid-tier creators?
Tier-2 uses private ID verification plus light friction. Tier-1 spontaneity stays intact. We'll add safeguards: quick one-time verification, grace periods/appeals, limited cooling-off, and exemptions for low-risk content.
> What makes Reddit exceptional?
Its multi-layer governance (admins, moderators, crowd votes) and the scale of volunteer moderation. We will emphasize this in the paper.
> Gaming reach thresholds (bots/spikes)?
We'll add anti-gaming measures such as anomaly detection, and coupling tier-gates to monetization/business-tool activation.

**3-5 Review Response3:**

7TUL

**3-6 Reaction To Review3:**

We thank the reviewer for the thoughtful and in-depth review. Especially the comment on our central claim being compelling: that tiered anonymity regulates amplification rather than censors speech. We'll emphasize this in the Introduction and add two clear ML touchpoints (recommender amplification; ML-assisted provenance/fact-checking). We'll also make Tier-3 ML-assisted pre-amplification review and provenance tagging (Table 1) more prominent.

To improve clarity, we'll add a brief "Definitions" section on pseudonymity vs. anonymity and proportionate friction, and change "restores" to "introduces" proportionate friction.

Regarding the Reddit case study, we already note that larger subreddits use stricter gates (karma minimums, approval queues) and sometimes identity proofs. We'll add a quantitative figure with a short methods note and point estimates, narrow the claim to "operational feasibility on large communities", note exceptions (identity-proof communities), include YouTube as a partial analogue, and soften the "difficult to replicate" phrasing with citations.

Regarding government vs. platform action: our aim is content-neutral amplification-gating and reach-tied duties implemented by platforms, consistent with our Section 230/First-Amendment analysis (Section 5.3). We'll add a footnote explaining the Section 230 "shield" and emphasise a platform-voluntary path via conditional immunity.

On global scope, we propose a common three-tier baseline with consistent semantics, paired with jurisdiction-specific thresholds and remedies, pursued through IGF/OECD-style soft law—addressing fragmentation and the "same tiers everywhere?" question.

We will also add the May 2025 "Take It Down Act" to Section 5.3 as a related development on traceability and identity processes - thank you for bringing this to our attention.

---

### Meta-Review · Area_Chair_sLvy · 2025-09-12

**Rating:** 6
**Confidence:** 5

**Strengths:**

In this paper, the authors argue that identity obligations should scale with influence and propose a tiered anonymity framework as a regulatory response. They clearly motivate the need for such a tiered approach to regulate amplification on social media platforms. The reviewers highlight the adequate grounding of the discussion in existing legal and regulatory contexts.

**Weaknesses:**

The main concerns include the lack of evidence supporting the claim that pseudonymity was preserved in the Reddit case, as well as the absence of a clear justification for why governments are the appropriate agents to implement the proposed framework. The reviewers also raise concerns on the risk of underestimation of enforcement challenges (e.g., legal, technical, or resistance by commercial platforms). Additionally, it does not sufficiently engage with the risks of misuse by authoritarian regimes for censorship purposes. The paper could also improve on clarity by defining key technical concepts (e.g., pseudonymity). It may also be valuable to consider alternative positions, such as the use of anonymous credentials, which could enable verification mechanisms without sacrificing user privacy.

**Questions:**

Reviewers raised questions about the need to ensure consistency of the tiered anonymity system across governments to avoid fragmentation, as well as the importance of implementing safeguards to prevent its misuse by authoritarian regimes.

**Ethics:**

No ethical violations or concerns were raised by the reviewers.

**Thoroughness:**

5

---

### Decision · Program_Chairs · 2025-09-26

Reject